# Validation of Malnutrition Screening Tools in Liver Cirrhosis

**DOI:** 10.3390/nu12051306

**Published:** 2020-05-03

**Authors:** Julia Traub, Ina Bergheim, Angela Horvath, Vanessa Stadlbauer

**Affiliations:** 1Department of Clinical Medical Nutrition, University Hospital Graz, 8036 Graz, Austria; julia.traub@klinikum-graz.at; 2Department of Nutritional Sciences, RF Molecular Nutritional Science, University Vienna, 1090 Vienna, Austria; ina.bergheim@univie.ac.at; 3Department of Gastroenterology and Hepatology, Medical University of Graz, 8036 Graz, Austria; angela.horvath@medunigraz.at

**Keywords:** malnutrition, liver cirrhosis, screening, assessment

## Abstract

Malnutrition in liver cirrhosis is frequently underestimated. To determine if a patient is at risk of malnutrition, several screening tools have been established. However, most of them are not validated for patients with liver cirrhosis. Therefore, we compared the RFH-NPT (Royal Free Hospital Nutritional Prioritizing Tool) as the validated gold standard for malnutrition screening in cirrhosis patients with GMS (Graz Malnutrition Screening), NRS-2002 (Nutritional Risk Screening) and MNA-SF (Mini Nutritional Assessment-Short Form). Based on common validity criteria for screening tools, only the MNA-SF showed fair correlation (12/15 points) with the RFH-NPT, whereas NRS-2002 and GMS performed worse (6/15 points). Taken together, our results suggest that NRS-2002 and GMS are not suitable for screening of malnutrition in cirrhosis patients. A cirrhosis-specific screening tool like RFH-NPT should be used to assess malnutrition and to identify those at risk of malnutrition.

## 1. Introduction

In patients with liver cirrhosis, hepatic metabolic functions are impaired resulting in a variety of nutritional disorders including protein-energy-malnutrition (PEM) which occurs in 65–90% but is still frequently underestimated [1]. However, up to now, common models to determine the prognosis in patients with liver cirrhosis do not include nutritional screening. Therefore, all liver cirrhosis patients should be additionally screened for the risk of malnutrition. Screening of malnutrition in liver cirrhosis is challenging because of the influence of fluid retention, such as ascites or peripheral oedema. Hence, conventional screening tools may not be valid [2]. The Royal Free Hospital Nutritional Prioritizing Tool (RFH-NPT), as a liver diseases specific screening tool, is recommended [3]. 

In clinical practice, malnutrition plays an important role for all hospitalized patients with different medical diagnoses, not only for patients with cirrhosis. In the current GLIM (Global Leadership Initiative on Malnutrition) criteria [4], the use of validated screening tools like the Nutritional Risk Screening-2002 (NRS-2002) or the Mini Nutritional Assessment-Short Form (MNA-SF) is recommended for all hospitalized patients. Unfortunately, fluid collections or disease severity are not included in these screening tools. The Medical University of Graz developed the Graz Malnutrition Screening (GMS), a validated screening tool, which includes different disease stages and severities like decompensated liver cirrhosis in the screening algorithm. However, the tool was not validated for cirrhosis patients separately [5]. 

In the present study, we aimed to validate three general malnutrition screening tools (MNA-SF, NRS-2002 and GMS) against RFH-NPT malnutrition in patients with liver cirrhosis.

## 2. Materials and Methods 

### 2.1. Study Design

This analysis is a sub-analysis of a prospective cohort study on sarcopenia in patients with liver cirrhosis. The trial was approved by the research ethics committee of the Medical University of Graz (29-280 ex 16/17) and was registered at clinicaltrials.gov (NCT03080129) before the start of recruitment. The study was conducted after written informed consent according to the principles of the Declaration of Helsinki. Hospitalized men or women over the age of 18 years with clinical/ radiological/ histological diagnosis of cirrhosis were included in the study. Exclusion criteria were defined as hepatic encephalopathy > grade 2 and/or other cognitive disorder not allowing for informed consent or hepatocellular carcinoma stage BCLC C or D. After written informed consent, RFH-NPT [3], GMS [5], NRS- 2002 [6] and MNA-SF [7] were assessed. GMS-Screening was performed routinely by the hospital staff and documented in the electronic patient records. An overview of the included screening elements of each screening tool is given in Table 1.

### 2.2. Statistics

Baseline characteristics were analyzed using descriptive statistics. Data are presented as frequencies, percentages and means with standard deviations (SD) for normally distributed variables. We performed statistical analysis by SPSS version 25 for Windows (IBM, Armonk, NY, USA). Sample size was based on feasibility with an average yearly rate of 70 patients with cirrhosis who were screened for malnutrition by the dietary team and an 80% consent rate. All eligible patients with complete datasets for all four nutritional screening tools were included; *p* < 0.05 was considered statistically significant.

The scores of each screening tool were normalized by scaling the rating values between 0 and 1. The normalized cut-off values for classifying malnutrition were calculated (RFH 0.286, GMS 0.300, NRS-2002 0.429, MNA-SF 0.214). Correlation between test results was assessed using Pearson’s correlation coefficient. Concurrent validity was determined by matching the RFH-NPT with the three other screening tools using the chi square test. Furthermore, sensitivity, specificity, positive and negative predictive values (PPV/NPV) and precision were calculated. For visualization of the differences between the various screening tools, Bland–Altman-Plot using differences versus averages were generated with a limit of agreement of 95%. Outliers, defined as values that fell outside the limits of agreement were defined. To evaluate the diagnostic accuracy area under the receiver operating curve (AUROC) was calculated. To determine predictive validity of the screening tools, the ratio between the odds ratios (ORs) and hazard ratios (HRs) were calculated. Kappa values were calculated for evaluating reliability. To rate validation results, published cut-offs were used. [8] For each test 3 points for good correlation, 2 points for fair correlation and 1 point for poor correlation was assigned and points were summed up. A higher score indicated a better validation result.

## 3. Results

### 3.1. Participant Characteristics

209 patients were screened for participation between March 2017 and March 2019. 84/209 did not meet inclusion criteria or fulfilled an exclusion criterion. In total, 125 patients were screened for the risk of malnutrition by four different screening tools, 7/125 were excluded because of incomplete data. Data from 118 patients were included in the analysis. Table 2 shows the baseline characteristics of the study population. 

### 3.2. Comparison between Screening Tools

In our study population, we saw the highest correlation between the RFH-NPT and the MNA-SF based on Person´s correlation (Pearson’s r values of 0.526). Chi-square for comparison of concurrent validity of screening tools was applied in all patients enrolled. Only GMS did not differ significantly from RFH-NPT (*p* = 0.163). Sensitivity was highest for MNA-SF (83%) compared to RFH-NPT, but precision was low (53%). GMS, NRS-2002 and MNA-SF had a high specificity (92%, 98%, 84%) but a very low sensitivity (39%, 22%) except for MNA- SF (83%). NPV were highest for MNA- SF (95%) and PPV for NRS-2002 (71%). Bland–Altman was calculated based on differences versus averages between RFH-NPT and MNA-SF, GMS as well as NRS-2002. Comparing bias, SD of bias and limits of agreement, MNA-SF showed the best agreement. The number of outliers was low for all comparisons. AUROC was calculated to evaluate the diagnostic accuracy and reliably between the different screening tools. The best AUROC was found between RFH-NPT and MNA-SF (0.834). To evaluate predictive validity of the screening tools, the ratio between ORs and HRs was calculated. Compared to RFH-NPT, MNA-SF and NRS-2002 were found to have the highest predictive value (2.159 and 2.932). MNA-SF had the highest Kappa coefficient (0.566) as measure for agreement in diagnosing malnutrition. An overview of the comparisons of the four screening tools is given in Table 3.

### 3.3. Validation of Screening Tools

The following validity criteria were chosen and weighted: Pearson correlation, sensitivity, specificity, AUROC, ORs and HRs, Kappa coefficient. Based on this rating, MNA-SF has the highest rating score (12/15, 80%, fair correlation) and therefore has the highest accordance to RFH-NPT. An overview of the rating is illustrated in Table 4.

## 4. Discussion

In our study, we validate three common general screening tools for malnutrition in liver cirrhosis by comparing them with the recommended liver disease specific screening tool (RFH-NPT). When using different statistical comparisons, MNA-SF has the highest rating score (12/15, 80%) and therefore the highest accordance to the RFH-NPT. MNA-SF gave false-negative results in comparison to RFH-NPT in only 17% whereas GMS and NRS-2002 have unacceptably high false-negative rates of 61% and 78%. False-positive results were obtained by MNA-SF in 16%, by GMS in 8% and by NRS-2002 in 2%. 

The requirements for a malnutrition screening tool are defined by the ESPEN guidelines for nutritional screening which should include the following four main principles: actual weight/BMI, recent weight loss, actual food intake and changes as well as disease process which accelerate nutritional deterioration [6]. The assessment of BMI is included in every screening algorithm but can be challenging in clinical routine because of the presence of fluid retention. Common general screening tools mostly do not take fluid overload into account; therefore, this can negatively influence the validity of the screening result.

The relevance of the diagnosis of malnutrition in patients with liver cirrhosis is highlighted in the new EASL Clinical Practical Guidelines for Nutrition in Chronical Liver Disease from 2019 [9], where a specific algorithm for nutrition screening in-patient with cirrhosis is suggested—the RFH-NPT. It is a well-validated screening tool, which considers fluid retention. Nutritional intervention in patients with cirrhosis who were found to be malnourished on screening improves nutritional status, presence of ascites and handgrip strength [10].

However, since malnutrition is common in all hospitalized patients [1], the use of malnutrition screening tools for the general hospital population is recommended [4]. A major challenge is to choose an appropriate screening tool that ideally fits all hospitalized patients and is easy to perform also by non-nutritional specialists. Such widely used tools are MNA-SF and NRS-2002 [2]. The Medical University of Graz developed the GMS which takes into account different diagnoses with an increased risk of malnutrition, such as the presence of decompensated cirrhosis [5]. However, the score did not perform better than MNA-SF and NRS-2002 compared to RFH-NPT in liver cirrhosis. This may be explained by the fact that GMS asks for decompensated cirrhosis only and not fluid retention and that the weight that is given to this diagnosis in the algorithm is not sufficient to represent the true risk of malnutrition. Furthermore, the NRS-2002 does not achieve satisfactory results for the screening for malnutrition in liver cirrhosis patients. Only the MNA-SF showed fair correlation compared to the RFH-NPT. 

## 5. Conclusions

Since in reality, only one screening tool that fits all hospitalized patients with different clinical diagnoses will be used, choosing a disease specific screening tool for live cirrhosis is not a useful option. Since none of the tested screening tools performed sufficiently in cirrhosis, we suggest the following workflow to improve the screening accuracy when using the electronic patient record system: A ‘stopping rule’ should be implemented into the workflow when a patient is diagnosed with liver cirrhosis. A message to the person doing the screening should be generated to inform that this screening tool is not validated for cirrhosis and that a dietician should be consulted.

## Figures and Tables

**Table 1 nutrients-12-01306-t001:** Summary of screening elements used in the nutrition screening tools.

Screening Elements	RFH-NPT	GMS	MNA-SF	NRS-2002
Alcoholic hepatitis or tube feeding	X			
Age		X		X
BMI		X	X	X
Dietary intake reduction	X			
Disease severity		X		X
Fluid overload	X			
Food intake		X		X
GI symptoms			X	
Mobility			X	
Neuropsychological problems			X	
Psychological stress/acute disease			X	
Weight loss	X	X	X	X

**Table 2 nutrients-12-01306-t002:** Baseline characteristics of patient population.

Characteristics	Study Population (*n* = 118)
Age (CI 95%)	65 (62.96–67.04)
Sex	
Men (%)	100 (84.7)
Woman (%)	18 (15.3)
BMI (CI 95%)	26.9 (25.99–27.81)
Etiology of cirrhosis	
HCV (%)	21 (17.8)
Alcohol (%)	70 (59.3)
NASH (%)	27 (22.9)
Child Pugh Score (CI 95%)	7 (6.61–7.39)
A	51
B	43
C	21
Meld Score (CI 95%)	10.9 (9.97–11.83)

**Table 3 nutrients-12-01306-t003:** Overview of criterion validity expressed by area under the receiver operating curve, odds ratios and hazard ratios, sensitivity, negative predictive values specificity, positive predictive values, precision, Pearson correlation and Kappa coefficient.

Validity Criteria	RFH-NPT-GMS	RFH-NPT-MNA-SF	RFH-NPT-NRS-2002
Pearson correlation	0.307	0.526	0.223
Sensitivity	39%	83%	22%
Negative predictive values	86%	95%	84%
Specificity	92%	84%	98%
Positive predictive values	53%	56%	71%
Precision	53%	56%	71%
Area under the receiver operating curve	0.654	0.834	0.598
Odds ratios and hazard ratios	1.831	2.159	2.932
Kappa coefficient	0.341	0.566	0.267

Abbreviations: RFH-NPT (Royal Free Hospital Nutritional Prioritizing Tool), GMS (Graz Malnutrition Screening), MNS-SF (Mini Nutrition Assessment-Short form), NRS-2002 (Nutritional Risk Screening).

**Table 4 nutrients-12-01306-t004:** Weighted results of screening validation.

Validity Criteria	RFH-NPT-GMS	RFH-NPT-MNA-SF	RFH-NPT-NRS-2002
Pearson correlation	1	2	1
Sensitivity & specificity	1	3	1
Area under the receiver operating curve	2	3	1
Odds ratios and hazard ratios	1	2	2
Kappa coefficient	1	2	1
Sum	6 ^1^	12 ^2^	6 ^1^
Relative Sum	40%	80%	40%

Good correlation, green; fair correlation, yellow; low correlation, red; ^1^ low correlation; ^2^ fair correlation [8], Abbreviations: RFH-NPT (Royal Free Hospital Nutritional Prioritizing Tool) and GMS (Graz Malnutrition Screening), MNS-SF (Mini Nutrition Assessment-Short form) and NRS-2002 (Nutritional Risk Screening).

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
