# Peer review of "Validation of Malnutrition Screening Tools in Liver Cirrhosis"

_nutrients, 2020, doi:10.3390/nu12051306_

Round 1
Reviewer 1 Report
Traub et al compared diagnostic accuracy of three general malnutrition screening tools
(MNA-SF, NRS-2002 and GMS) with RFH-NPT for screening of malnutrition in cirrhosis patients. The authors found that the MNA-SF exclusively shows fair correlation compared with the RFH-NPT. However, this paper has a number of short comings, details of which are given below.
Concerns
- The number of patients were too small to validate the tool for detecting malnutrition in patients with liver cirrhosis.
- The authors should collected two independent dataset to evaluate the diagnostic accuracy of malnutrition screening tools .
- The authors should show the baseline characteristics of the patients with cirrhosis.
- I would encourage the authors to proofread the paper carefully and have it checked by a native speaker of English.
Author Response
We thank the reviewer for his/her favourable comments on our manuscript.
- The number of patients were too small to validate the tool for detecting malnutrition in patients with liver cirrhosis.
We agree with the reviewer, that a higher number of patients is always desirable for validating studies. Recent comparable validation studies were performed with patient numbers between 80 to 170. [Borhofen, Sarah Maria, et al. Digestive diseases and sciences 61.6 (2016): 1735-1743; Georgiou, Alexandra, et al. British Journal of Nutrition 122.12 (2019): 1368-1376; Morgan, Marsha Y., et al. Hepatology 44.4 (2006): 823-835.] Therefore, we believe that the patient number is sufficient for this short report.
But we take up this suggestion and plan to validate our results further in a larger patient cohort.
- The authors should collected two independent dataset to evaluate the diagnostic accuracy of malnutrition screening tools.
We agree with the reviewer, that an independent validation dataset would enhance the impact of our results. Currently it is unfortunately not possible at our university or at any of our partner institutions to start such a study due to the Corona pandemic and the associated restrictions to perform studies that are not directly related to Covid-19. We aim to set up such a study in the future. The collection of a comparable dataset will take about 1-2 years, depending on the volume of patients with liver cirrhosis in an institution.
- The authors should show the baseline characteristics of the patients with cirrhosis.
Thank you for this valuable comment. We added a table with baseline characteristics of the study population.
- I would encourage the authors to proofread the paper carefully and have it checked by a native speaker of English.
We proofread the paper carefully ourselves and had the paper proofread by an English native speaker who offers professional proofreading service.
Reviewer 2 Report
The authors have performed a study to compare different methods for nutritional assessment in patients with advanced liver disease.
I have some comments which may improve the manuscript.
Page 2 line 55-57;- it is always disappointing for the reader to need to go to other references to understand the methods. I think it is worth including a small Table showing at least the parameters included in the nutritional scores.
Page 2 Lines 62-63: The sentence is not correct. A sample-size calculation is always needed also in observational studies. This is to be sure that the sample is adequate for the statistical power of the analysis.
Discussion: please add in a few lines the number of misdiagnoses using the different methods. This will give also a clinical dimension of the results.
Author Response
We thank the reviewer for his/her favourable comments on our manuscript.
- Page 2 line 55-57;- it is always disappointing for the reader to need to go to other references to understand the methods. I think it is worth including a small Table showing at least the parameters included in the nutritional scores
We added a table that summarizes the screening elements of each screening tool.
- Page 2 Lines 62-63: The sentence is not correct. A sample-size calculation is always needed also in observational studies. This is to be sure that the sample is adequate for the statistical power of the analysis.
We agree with the reviewer, that the sentence is not correct- the sentence was deleted and replaced by the sentence: “Sample size was based on feasibility with an average yearly rate of 70 patients with cirrhosis who were screened for malnutrition by the dietary team and an 80% consent rate.”
- Discussion: please add in a few lines the number of misdiagnoses using the different methods. This will give also a clinical dimension of the results.
Thank you for this valuable comment. We added the following statement to the discussion “MNA-SF gave false-negative results in comparison to RFH-NPT in only 17% whereas GMS and NRS-2002 have unacceptably high false-negative rates of 61% and 78%. False-positive results were obtained by MNA-SF in 16%, by GMS in 8% and by NRS-2002 in 2%.“
Reviewer 3 Report
In this study, Traub et al. validate different screening tools to detect malnutrition in cirrhotic patients. They found that only the Mini Nutritional Assessment-Short Form (MNA-SF) showed a good correlation with the Royal Free Hospital Nutritional Prioritizing Tool as the validated gold standard.
Overall, this is a very carefully performed study and I don`t have major criticism.
Minor points:
-In the manuscript “women” is misspelled as “woman”
-It should be discussed how screening for malnutrition with consecutive dietary interventions affects outcome in cirrhotic patients, is there any data available? This would underline the importance of the implementation for malnutrition screening.
Author Response
We thank the reviewer for his/her favourable comments on our manuscript.
- In the manuscript “women” is misspelled as “woman”
“Woman” was changed to “women”.
- It should be discussed how screening for malnutrition with consecutive dietary interventions affects outcome in cirrhotic patients, is there any data available? This would underline the importance of the implementation for malnutrition screening
We agree with the reviewer, that the effect of screening of malnutrition and the consecutive dietary intervention on outcome data needs to be mentioned. Therefore, we added the following information: „Nutritional intervention in patients with cirrhosis who were found to be malnourished on screening improves nutritional status, presence of ascites and handgrip strength. [Vidot, Helen, et al. JGH Open 1.3 (2017): 92-97.].”
Round 2
Reviewer 1 Report
I have no further concerns